# Insertion of ammonia into alkenes to build aromatic *N*-heterocycles

Shuai Liu[1,2] & Xu Cheng [1,3 ✉]

Ammonia is one of the most abundant and simple nitrogen sources with decent stability and reactivity. Direct insertion of ammonia into a carbon skeleton is an ideal approach to building valuable *N*-heterocycles for extensive applications with unprecedented atom and step economy. Here, we show an electrochemical dehydrogenative method in which ammonia is inserted directly into alkenes to build aromatic *N*-heterocycles in a single step without the use of any external oxidant. This new approach achieves 98–99.2% atom economy with hydrogen as the only byproduct. Quinoline and pyridine with diverse substitutions are readily available. In this work, electrochemistry was used to drive a 4-electron oxidation reaction that is hard to access by other protocols, providing a parallel pathway to nitrene chemistry. In a tandem transformation that included three distinct electrochemical processes, the insertion of ammonia further showcased the tremendous potential to manipulate heterocycles derived from Hantzsch ester to diazine via pyridine and pyrrole.

[1] Institute of Chemistry and Biomedical Sciences, Jiangsu Key Laboratory of Advanced Organic Materials, School of Chemistry and Chemical Engineering, National Demonstration Center for Experimental Chemistry Education, Nanjing University, Nanjing 210023, China. [2] School of Materials and Chemical Engineering, Xuzhou University of Technology, Xuzhou 221018, China. [3] State Key Laboratory of Elemento-organic Chemistry, Nankai University, Tianjin 300071, China. ✉email: chengxu@nju.edu.cn

Aromatic *N*-heterocycles are common moieties in natural products[1], and they have been extensively applied to pharmaceutical design[2]. The classic syntheses of aromatic *N*-heterocycles are frequently conducted with condensation reactions using pre-oxidized materials. For example, in early efforts to synthesize isoquinoline from indene, oxidation cleavage of alkenes using ozone[3,4] or $OsO_4$[5] was necessary, resulting in a limited scope of functionality (Fig. 1a)[6]. However, the insertion of heteroatoms into the carbon skeleton exhibits multiple advantages, including atom/step economy, oxidation-labile functional group (Fg) tolerance, and unexplored selectivity (Fig. 1b). However, it is difficult to find a method as efficient as oxidation for cleaving C–C bonds, especially the bonds of alkenes and alkynes. Such an intrinsic obstacle makes the direct insertion of heteroatoms into alkenes to build heterocycles an elusive goal. To regulate the oxidation state of organic molecules, the dehydrogenation reaction is a complementary route to the oxygenation reaction[7–21]. Dehydrogenative cross-coupling exhibits a unique ability to achieve oxidative bond formation in the absence of external oxidants[22–26]. In particular, the electrochemical protocol demonstrates its intrinsic potential to drive dehydrogenative cross-coupling reactions with cathodic hydrogen evolution[27–30]. As one of the most important C-heteroatom connections, considerable efforts have been made to construct C-N bonds, including electrochemical protocols[31–33]. In recent years, the electrochemical dehydrogenative intermolecular construction of C-N bonds has witnessed tremendous progress, for example, in reactions of aromatic C-H amination[34–43], benzylic C-H amination[44–46], alkyne amination[47], alkene azidation[48], alkane amination[49], aromatic C-X/N-H cross coupling[50–53], alkene aziridination[49,54–59], and other innovations[60]. Despite these achievements, electricity has not fulfilled the task of sewing $NH_3$ and alkene to aromatic *N*-heterocycles.

In this work, we report the insertion of ammonia into carbon skeletons to build diverse aromatic *N*-heterocycles via a multiple electron transfer pathway involving hydrogen evolution with up to 99.2% theoretical atom economy.

## Results

**Optimization of insertion of ammonia into indene.** In the initial attempt, **1a** was adopted as the substrate with the conjugated tetrasubstituted alkene as the target bond to insert a nitrogen atom with ammonia (Table 1) at 0 °C. The first observed insertion product **2a** had a 16% [1]H NMR yield when graphite felt (GF) was used as the electrodes and methanol was used as the solvent (entry 1). Pt (entry 2) and Ag (entry 3) were applied as cathodes to enhance the evolution of hydrogen, and the

**Table 1 Initial model reaction of ammonia insertion and optimization.**

| Entry | Solvent | Electrodes | Electrolyte | Yield (%)[g] |
|---|---|---|---|---|
| 1[a] | $CH_3OH$ | GF+/GF− | $Mg(ClO_4)_2$ | 16 |
| 2[a] | $CH_3OH$ | GF+/Pt− | $Mg(ClO_4)_2$ | 30 |
| 3[a] | $CH_3OH$ | GF+/Ag− | $Mg(ClO_4)_2$ | 52 |
| 4[b] | $CH_3OH$ | GF+/Ag− | $Mg(ClO_4)_2$ | 50 |
| 5[c] | $CH_3OH$ | GF+/Ag− | $Mg(ClO_4)_2$ | 47 |
| 6[d] | $CH_3OH$ | GF+/Ag− | $Mg(ClO_4)_2$ | Trace |
| 7[e] | $CH_3OH$ | GF+/Ag− | $Mg(ClO_4)_2$ | 55 |
| 8[f] | $CH_3OH/DCM$ | GF+/Ag− | $Mg(ClO_4)_2$ | 68(65[h]) |
| 9[i] | $^iPrOH/DCM$ | GF+/Ag− | $Mg(ClO_4)_2$ | N. R. |
| 10[j] | $CH_3OH/DCM$ | GF+/Ag− | $Mg(ClO_4)_2$ | 64 |
| 11[f] | $CH_3OH/DCM$ | GF+/Ag− | LiCl | 30 |
| 12[f] | $CH_3OH/DCM$ | GF+/Ag− | $LiBF_4$ | 67 |
| 13 | $NH_3/Pb(OAc)_4$ | | | N.R. |

[a] Reaction conditions: **1a** (0.1 mmol), $NH_3$ (balloon, ca. 1 atm), graphite felt (GF) anode, Ag cathode, supporting electrolyte (0.1 mmol), MeOH (5.0 mL), 0 °C, 4 V cell voltage, 3 h.
[b] 3.5 V cell voltage.
[c] 5 V cell voltage.
[d] −10 °C.
[e] rt instead of 0 °C.
[f] MeOH (4.0 mL) and DCM (dichloromethane, 1.0 mL), rt, 3 h.
[g] [1]H NMR yields of product **2a**.
[h] Isolated yields of product **2a**.
[i] $^iPrOH$ (4.0 mL) and dichloromethane, DCM (1.0 mL), rt, 3 h.
[j] $NH_3$ (0.28 mol/L).

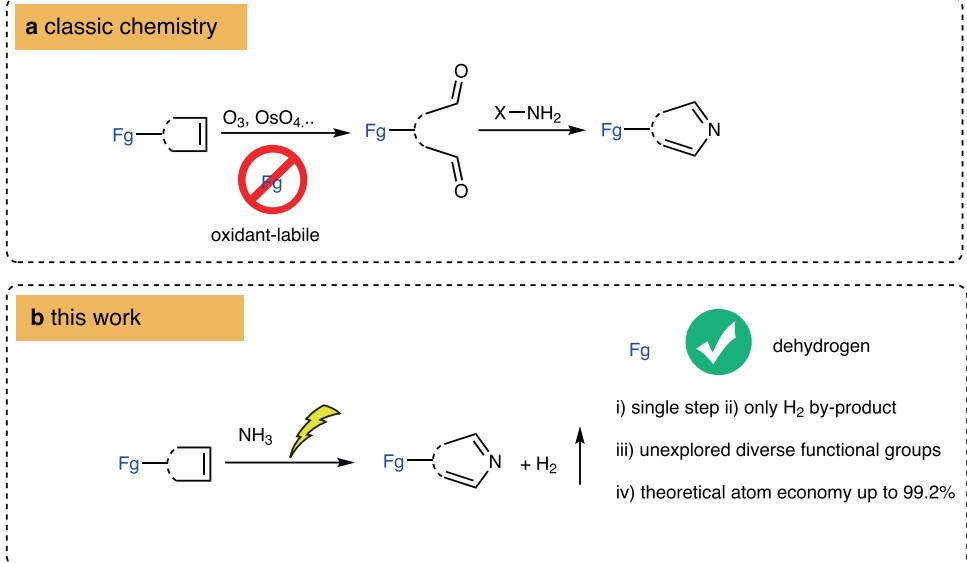

**Fig. 1 Comparison of protocols to construct aromatic *N*-heterocycles from alkenes. a** Classic chemistry, oxidation/condensation. **b** This work, dehydrogenative insertion of $NH_3$. Fg = functional groups.

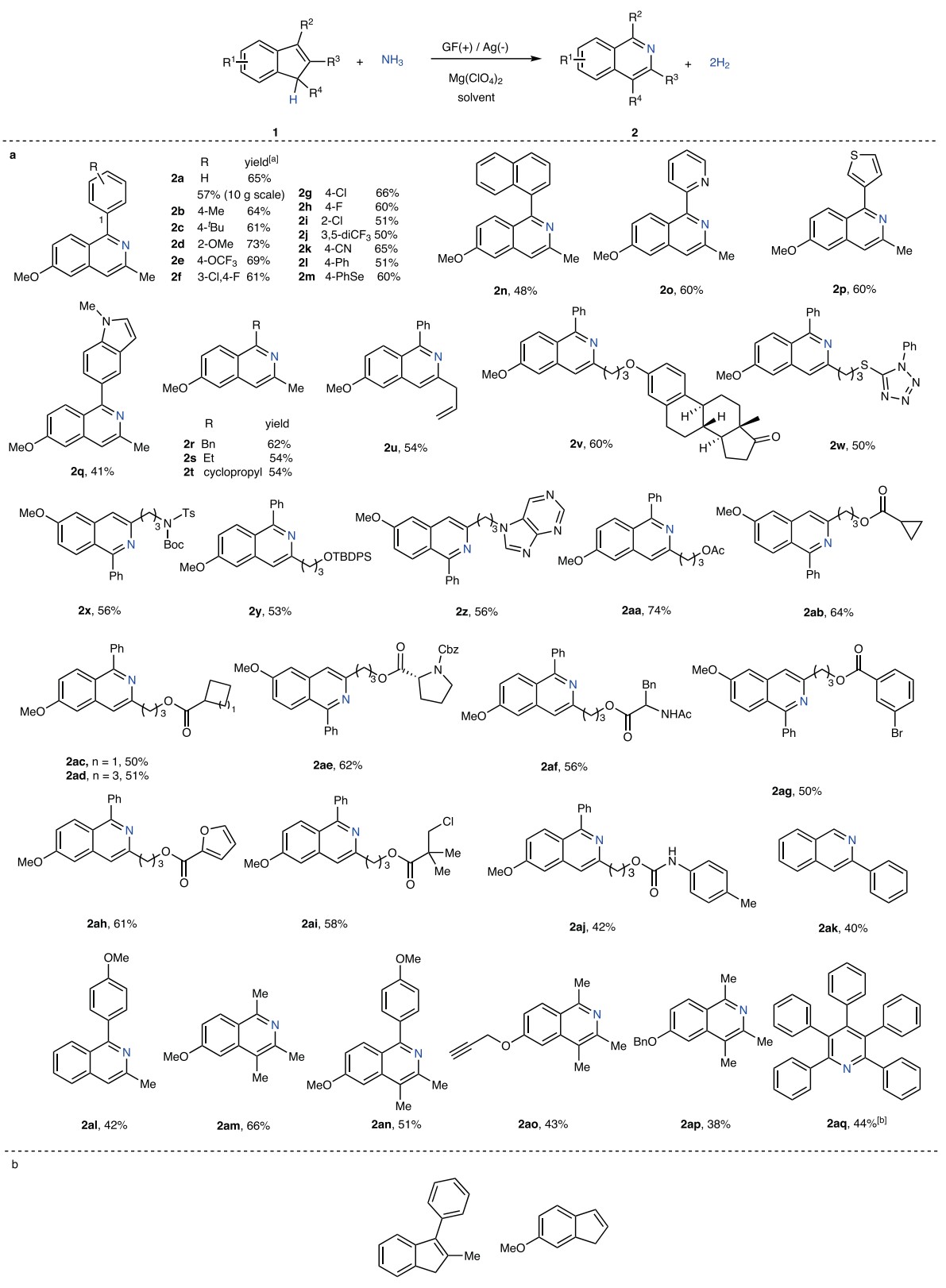

**Fig. 2 The insertion of ammonia into various alkenes. a** Reaction conditions: graphite felt (GF) anode and Ag cathode, **1** (0.1 mmol), NH₃ (balloon, ca. 1 atm), Mg(ClO₄)₂ (0.1 mmol), MeOH/DCM = 4 mL/1 mL, or 3 mL/2 mL, 3.5–4.5 V cell potential, rt, 3–4 h. [a] isolated yields are reported for all cases. [b] reaction conditions: GF anode and Ag cathode, **1aq** (0.05 mmol), Mg(ClO₄)₂ (0.1 mmol), MeOH/toluene = 3 mL/2 mL, 5.5 V cell potential, rt, 3 h. **b** Substrates did not react.

**Fig. 3 The application of ammonia insertion protocol. a** Three-step electrochemical synthesis of diazine **2ar** from Hantzsch ester **3**. **b** Synthesis of choroidal blood regulator moxaverine **2as**. **c** Synthesis of 6-methoxy-3-methyl-[1,1'-biisoquinoline] 2,2'-dioxide. GF, graphite felt; mCPBA; LDA, lithium diisopropylamide; 3-chloroperoxybenzoic acid, PMP, 4-MeOPh.

corresponding yields were improved to 30 and 52%, respectively. The decrease (entry 4) and increase (entry 5) in cell voltage slightly decreased the yield. Lowering the reaction temperature to $-10\,°C$ brought only trace conversion (entry 6). However, the reaction at room temperature could give the desired isoquinoline in comparable yield (entry 7). The application of DCM as a co-solvent could improve the solubility of the substrate and lead to a 65% isolated yield (entry 8). When $^iPrOH$ was the solvent with dichloromethane (DCM) as the co-solvent, the conversion of alkene was not observed due to the low conductivity (entry 9). Instead of an ammonia atmosphere, a solution of ammonia at a concentration of 0.28 mol/L was adequate to give a comparable yield (entry 10). The screening of other supporting electrolytes showed that LiCl resulted in inferior yields (entry 11), and LiBF$_4$ gave acceptable results (entry 12). A reaction using Pb(OAc)$_4$ as the terminal oxidant[61] with ammonia did not give conversion of the substrate (entry 13, see Supplementary Tables 1 and 2 for more details).

**Substrate scope of reaction of insertion of ammonia**. With the optimized conditions (Table 1, entry 8), we explored the scope of

the other cyclic alkenes (Fig. 2). Products **2a–2 m** with different aryl groups at the R$^2$ position were obtained in moderate to good yields. In a 10-gram scale reaction, product **2a** was isolated in a similar yield (57%) to that shown in Table 1 (65%, entry 6). Next, naphthalenyl **2n** and biheterocycles **2o–2q** were prepared with the same protocol. The electron-deficient pyridine **2o** and electron-rich thiophene **2p** and indole **2q** were all compatible. Next, the reactions employing substrates bearing alkyl R$^2$ groups gave the products **2r–2t**. In the case of **2t**, the radical clock cyclopropyl group was intact during the transformation. Next, the other R$^3$ side chains were evaluated using a series of functional groups. Allyl **2 u**, estrone-derived ether **2 v**, mercaptophenyltetrazole-derived thioether **2w**, amide protected with $^t$butoxycarbonyl (Boc) and p-tolyl sulfonyl (Ts) **2x**, silyl ether **2 y**, and purine **2z** were all robust under these reaction conditions, and the corresponding products were achieved in 50–60% yield. In addition, substrates **1**, which incorporated different ester groups, were synthesized as tested under the standard conditions, and the desired products **2aa–2ai** were obtained in up to 74% yield, revealing further tolerance towards Ar-Br **2ag**, furan **2ah**, and R-Cl **2ai** groups. A carbamate **2aj** was achieved in moderate 42% yield. Next, the

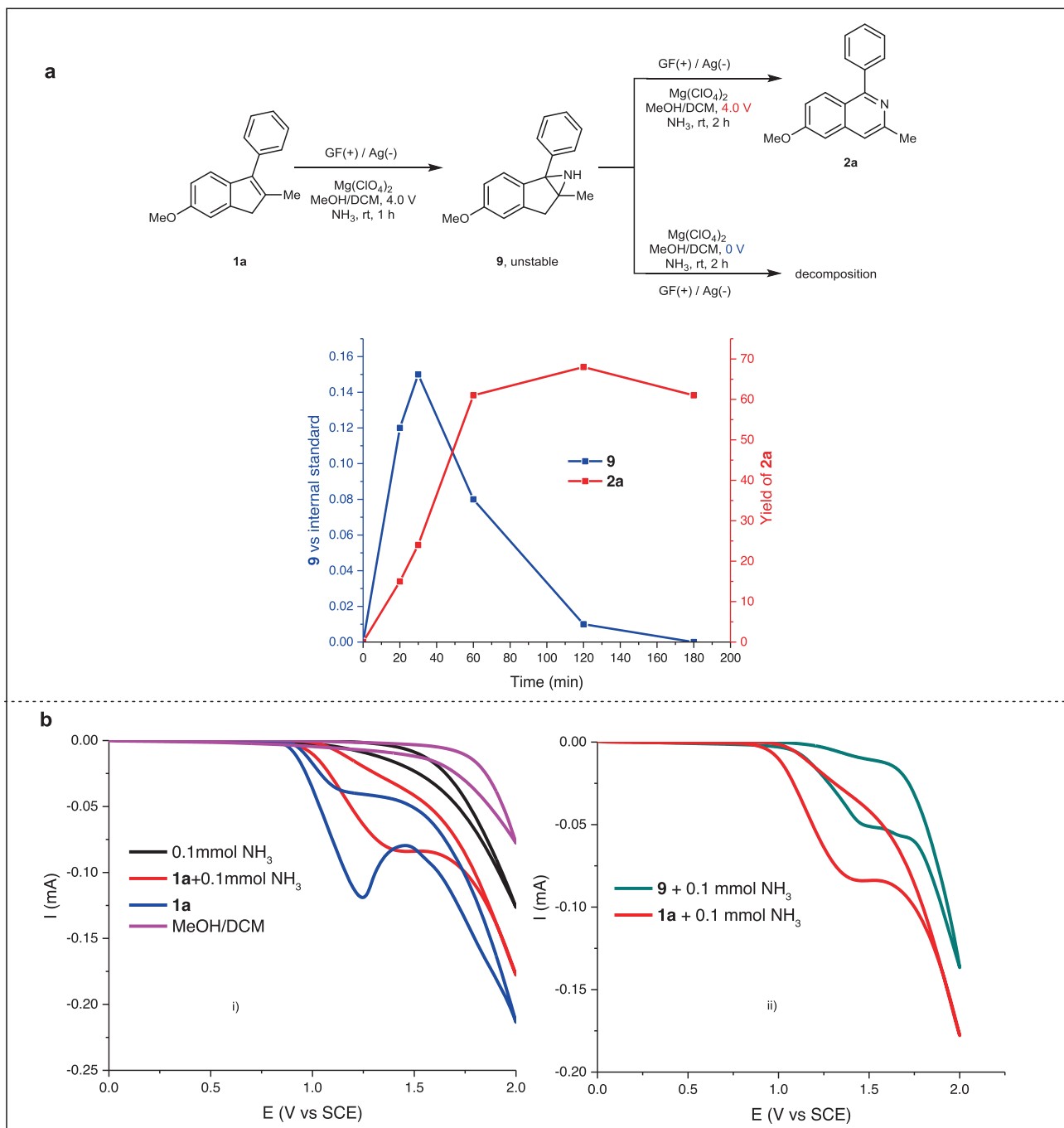

**Fig. 4 Experiments investigating the reagents and intermediates. a** Investigation and tracking of intermediates **9** under different conditions. **b** Cyclic voltammetry experiments of reactants and **9**. GF, graphite felt; DCM, dichloromethane.

indenes **1** with varied substitution patterns was converted to iso-quinolines **2ak–2ap**. In these cases, the yields were lower partially because of the unidentified side reactions involving MeOH as a nucleophile. We further explored the substrate scope beyond the indene. Tetraphenyl cyclopentadiene was converted to the corresponding tetraphenyl pyridine **2aq** in a single step in 44% yield. Functional groups, such as alkenes, alkynes, selenides, thioethers, thiophenes[62,63], and indole[64], which react with ozone readily, were all tolerated during the transformations. In the example of **2 v**, a theoretical atom economy of 99.2% was realized. However, there was a limit on the pattern of substitution. For example, substrates **1au** and **1av** showed that the electron-donating group on the

indene backbone and aryl ring on the alkene were important because they were capable of stabilizing the cationic and radical intermediates.

**Tandem reaction involving the insertion of ammonia**. Subsequently, we explored the insertion of ammonia into alkenes in a three-step tandem electrochemical reaction (Fig. 3a). By employing Hantzsch ester **3** as the starting material, tandem electrochemical dehydrogenative oxidation/reductive ring contraction/NH₃ insertion gave 1,3-diazine **2ar** without isolation of intermediates. The overall side products for this transformation were ethyl acetate[65] and hydrogen, and the obtained tetra-

**Fig. 5 A plausible reaction pathway involving 4e oxidation. a** Anodic reactions converting substrate **1** to product **2**, GF( + ) is graphite felt anode. **b** Cathodic hydrogen evolution.

substituted 1,3-diazine was not accessible by any other methods (see Supplementary Figs. 36 and 37 for more details). With this method, the commercial pharmaceutical compound moxaverine **2as** was synthesized in four steps from indanone **4** with ammonia insertion as the key step in 62% yield (Fig. 3b). Next, Compound **7** was converted to unsymmetric bi-quinoline **2at**, which was further oxidized to N-O oxide with 3-chloroperoxybenzoic acid (mCPBA). The achieved bi-N-O oxide **8** was an axial chiral scaffold (Fig. 3c, see Supplementary Fig. 38 for more details)[66].

**Mechanistic studies**. Next, experiments under controlled conditions were conducted to gain some information about the reaction pathway (Fig. 4). At first, the reaction of **1a** and ammonia was interrupted at an early stage before the full consumption of the starting alkene. An unstable intermediate aziridine **9** was detected, isolated, and characterized. Next, we subjected Compound **9** to the same electrochemical conditions, and the final isoquinoline **2a** was produced as the major product. However, when Compound **9** was subjected to the same chemical environment without electricity, decomposition to an unknown complex mixture was observed, and target Compound **2a** was not detected at all (see Supplementary Figs. 7–9 for more details). In the kinetic study, the concentration of intermediate **9** reached a maximum at 30 min and then decreased to full consumption until 180 min. In comparison, product **2a** was generated continuously for 120 min, and then the yield dropped slightly (Fig. 4a, see Supplementary Figs. 10 and 11 for more details). Next, cyclic voltammetry (CV) was performed on the species in the reaction (Fig. 4b). The results revealed that **1a** more readily lost electrons than ammonia when it was the sole solute in MeOH (Fig. 4b-i, blue curve). When **1a** and ammonia were exposed to anodic oxidation together, the onset potential (1.0 V vs. saturated calomel electrode SCE) of the reactants was similar to that of **1a**, but the peak potential shifted to 1.5 V vs. SCE, suggesting that ammonia affected the electrode environment during electron transfer from **1a** to the anode (Fig. 4b-i, red curve). Furthermore, it was found that the oxidation of **1a** proceeded slightly more readily than the oxidation of the intermediate (**9**) (Fig. 4b-ii, see Supplementary Figs. 21–26 for details).

**A plausible reaction mechanism**. With these observations, we suggest a plausible reaction pathway, shown in Fig. 5. First, the reaction starts with anodic oxidation of alkene **1** and the subsequent trapping of cationic radical **A** with ammonia (for more details of the radical kinetics, see Supplementary Figs. 15–19). The second oxidation converts the consequent neutral radical **B** to cation **C**, which undergoes annulation to aziridine **D**. A third electron transfer oxidation of nitrogen during the conversion of **D** to **E** triggers deprotonation/rearrangement, yielding dihydroisoquinoline radical **F**. The fourth electron transfer and deprotonation results in the final product **2**. Finally, the evolution of **2** equivalents of hydrogen molecules at the cathode accomplishes the whole electron cycle.

## Discussion

In summary, we developed a direct insertion of ammonia into cyclic alkenes to synthesize aromatic N-heterocycles. The reaction utilizes electrochemical hydrogen evolution instead of oxygenation to drive the reaction. The reaction proceeds via electrochemical aziridination and ring rearrangement with up to 99.2% theoretical atom economy. By avoiding the usage of external oxidants, various functional groups labile towards oxidation were compatible in this transformation. Divergent N-heterocycles are available with this oxidant-free pathway.

## Methods

**General procedure for the synthesis of 2**. A 10 mL three-necked heart-shaped flask was charged with the substrate alkene (0.1 mmol), Mg(ClO$_4$)$_2$ (22.3 mg, 0.1 mmol) and a magnetic stir bar. The flask was equipped with a rubber stopper, graphite felt (2 cm × 1 cm x 0.5 cm) as the anode and Ag plate (2 cm × 1 cm) as the cathode. The flask was evacuated and backfilled with ammonia gas three times, and then an ammonia gas balloon was connected to this flask via a needle. Next, anhydrous solvent (5 mL) was added via syringe. Electrolysis with constant cell potential was carried out at room temperature. The reaction was monitored with TLC and GC–MS, and when it was complete, the mixture was concentrated under reduced pressure. The residue was purified by chromatography on silica gel to afford the desired product (see Supplementary Fig. 1 for more details).

Gram-scale reaction to prepare **2a**: A 400 mL rectangular flask was charged with substrate 1a (10.0 g, 42 mmol), LiBF$_4$ (0.60 g, 6 mmol) and a magnetic stir bar. The flask was equipped with two pieces of graphite felt (8.5 cm × 6.5 cm × 0.5 cm, 6.5 cm × 6.5 cm × 0.5 cm) as the anode (2 pieces) and one piece of silver flake (6.5 cm × 6.5 cm × 0.5 cm) as the cathode. Two electrodes were separated and fixed

with a 1.0 cm stick. The graphite felt anode was attached to a platinum wire, and cathode was attached to a silver wire (see Supplementary Figs. 2–6 for details). The flask was evacuated once and backfilled with gaseous $NH_3$, and 200 mL of anhydrous MeOH and 100 mL of anhydrous DCM were added via syringe. Electrolysis under a controlled cell potential (8 V due to the extended distance between anode and cathode) was carried out in a water bath at room temperature. After 14 h, the mixture was concentrated under reduced pressure. The residue was purified by chromatography on silica gel to afford desired product **2a** (6.0 g, 57%).

## Data availability

The authors declare that all other data supporting the findings of this study are available within the article and Supplementary Information files and are also available from the corresponding author upon request.

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

## Acknowledgements

This work was supported by the National Science Foundation of China (nos. 22071105, 22031008, Xu Cheng) and the Qinglan Project of Jiangsu Education Department (Xu Cheng).

## Author contributions

S.L. carried out the work on optimization of the reaction conditions, exploration of the substrate scope, elucidation of the mechanism and wrote the supplementary information. X.C. conceived and supervised the project and wrote the manuscript.

## Competing interests

The authors declare no competing interests.
