## [Peer Review File · Nature Communications]

Insertion of Ammonia into Alkenes to Build Aromatic N-HeterocyclesREVIEWER COMMENTS

Reviewer #1 (Remarks to the Author):

Cheng and Liu report an interesting entry of synthesis of 6-membered N-heterocycles by oxidative coupling of indene and cyclopentadiene derivatives with ammonia under electrochemical oxidation conditions. They examined the reaction using various cyclic dienes and obtained a variety of 6-membered N-heterocycles in moderate to good yields. The electrochemical oxidation reaction participates in the generation of aziridines and oxidative rearrangement of the aziridines. Narasimhan's group reported a similar result using cyclopentadienes, N-aminophthalimide as a nitrogen source, and Pb(OAc)₄ as an oxidant (Heterocycles 1984, 22, 1369). Cheng only changed Pb(OAc)₄ to electrochemical oxidation. As they claimed, these authors' reaction system using electrochemical oxidation is clean and atom-economical, but the novelty of the transformation, which is the most crucial point for publication in Nature Communications, does not satisfy this journal's requirement. In addition, the yields of the products are insufficient (less than 60%) in many cases. And there is no discussion regarding the relationship between the substituents and the reactivities. They proposed the generation of benzyl radical intermediate B in Figure 5, but the cyclopropyl group used as a radical clock remained intact in product 2t. They did not explain this inconsistent result.

As mentioned above, this reviewer does not recommend publication their result in Nature Communications.

Reviewer #2 (Remarks to the Author):

Liu and co-workers reported an impressive synthesis of aromatic N-heterocycles. The utilization of abundant, stable, and nitrogen-enriched ammonia directly and selectively in organic synthesis is a long-sought target, and has been shown as a highly challenging task. The activation of ammonia to nitrene or amide radical is the focused direction. Instead, this report took the other route with activation of alkene in the presence of ammonia. This transformation exploited the unique advantages of electrochemistry, such as non-acidic activation of alkene under ammonia conditions, four electron transfers in quick sequence which is so hard for other methods, and the chemoselectivity achieved by anodic oxidation converting aziridine. The protocol approaches the quantitative limit of atom economy, and meanwhile achieves a broad functional tolerance and diverse products scope. Some interesting and valuable molecules were generated in single step or with multiple electrochemical process. Overall, this research is particularly noteworthy. So, it deserves my support to publish on Nature Communications.

There were several comments on the manuscript and supporting information:

- 1) As the authors say, the lower yields (2ak–2ap) were partially because of the unidentified side reactions involving MeOH as nucleophile. Can sterically hindered alcohol (such as isopropyl alcohol) instead of MeOH improve the reaction?
- 2) NH₃ gas was directly introduced into the reaction. Does the concentration of NH₃ affect the reaction efficiency?
- 3) Aqueous ammonia would be readily to handle and store. How about the result of NH₄OH in this reaction?
- 4) The discussion on the synthesis of 2ar need additional information.

Reviewer #3 (Remarks to the Author):

In this communication, Chen and their co-worker reported a rare example to construct aromatic N-heterocycles via insertion of ammonia. This process was enabled via electrochemical aziridination and ring expansion. This operationally straightforward proved useful for the preparation of versatile heterocycles and pharmaceutical compounds in good yields. The results are promising, and the authors showcase the potential for further applications. This manuscript has its own merit in the employment of electrosynthesis, demonstrating various aromatic N-heterocycles with good yields in a concise manner, and this work will catch the attention of a broad readership. To sum up, I think that this is a nice piece of work well suited to be published in Nature Communication after revising the items noted below:

- 1) The ammonia gas pressure used in the reaction should be added to the table footnote.
- 2) Given the full name of Fg and GF.
- 3) ¹H NMR of 1m was not clean and should be improved.
- 4) Kinetic studies of intermediate aziridine 9 and the product were suggested to be investigated.
- 5) In Fig. 3a, a nice electrochemical sequence for the synthesis of 1,3-diazine 2ar was reported. The intermediates should also be added to the manuscript. Furthermore, could substitute pyrrole be the suitable substrate in the transformation?

REVIEWER COMMENTS

Reviewer #1 (Remarks to the Author):

Cheng and Liu report an interesting entry of synthesis of 6-membered N-heterocycles by oxidative coupling of indene and cyclopentadiene derivatives with ammonia under electrochemical oxidation conditions. They examined the reaction using various cyclic dienes and obtained a variety of 6-membered N-heterocycles in moderate to good yields. The electrochemical oxidation reaction participates in the generation of aziridines and oxidative rearrangement of the aziridines.

Comment:

Narasimhan's group reported a similar result using cyclopentadienes, *N*-aminophthalimide as a nitrogen source, and Pb(OAc)₄ as an oxidant (*Heterocycles* **1984**, 22, 1369). Cheng only changed Pb(OAc)₄ to electrochemical oxidation. As they claimed, these authors' reaction system using electrochemical oxidation is clean and atom-economical, but the novelty of the transformation, which is the most crucial point for publication in Nature Communications, does not satisfy this journal's requirement.

Response:

Thanks a lot for this important comment. As that suggested in the title of *Heterocycle* **1984**, 22, 1369, "Addition of Phthalimido Nitrene to Substituted Cyclopentadienes", the nitrene is the key intermediate in the transformation, which have been well-studied. The substitution phthalimide on the amino group could stabilize the nitrene intermediate. In comparison, the ammonia is difficult to turn to nitrene. As that suggested in the computation study¹ and mass spectroscopy study², to date, nitrene from ammonia only existed in gas phase under high vacuum.

1. Buckner SW, Gord JR, Freiser BS. Gas-phase chemistry of transition metal-imido and -nitrene ion complexes. Oxidative addition of nitrogen-hydrogen bonds in ammonia and transfer of NH from a metal center to an alkene. *J. Am. Chem. Soc.* **110**, 6606-6612 (1988).
2. Kretschmer R, Wang Z-C, Schlangen M, Schwarz H. Single and Double N-H Bond Activation of Ammonia by [Al₂O₃]⁺: Room Temperature Formation of the Aminyl Radical and Nitrene. *Angew. Chem. Int. Ed.* **52**, 9513-9517 (2013).

To the best of our knowledge, the existence of ammonia derived nitrene in solution is still unexplored. This work exhibited another pathway utilizing ammonia in solution. The ammonia works as nucleophile twice in the aziridination reaction.

To test the chemistry of Pb(OAc)₄, a reaction using Pd(OAc)₄ as terminal oxidant instead of electricity with ammonia was carried out. This reaction did not give the conversion of substrate alkene.

This result was added in the manuscript, Table 1, entry 13.

A reaction using Pb(OAc)₄ as terminal oxidant⁶¹ with ammonia did not give conversion of

Comment:

In addition, the yields of the products are insufficient (less than 60%) in many cases.

Response:

Thanks a lot for this comment. We checked our reaction again and found several reasons for the low yields.

- 1) The crude ¹H NMR with internal standard showed the yield is normally 3-10% higher than that from separation after silica gel chromatography. Especially, for substrate bearing more than one N atom.
- 2) There were several functional group, for example, in product **2q**, **2ao**, **2ap**, which are not strong enough to be completely intact in this transformation. The decomposition of these functional groups led to low yields.
- 3) Some product suffered the poor solubility. For example, **2n** and **2aq** are poorly soluble during workup and purification, which lead to the loss of yield during these stages.
- 4) There were some side products along with the target compounds. From the crude ¹H NMR of product **2a**, it was found adduct of MeOH estimated to be about 10%.

¹H NMR analysis of crude mixture of standard reaction

Prompt by this comment, we tried to carry out the reaction and isolation again for some substrate with low yields. Some improvements of yields were achieved. These results were added in Figure 2.

Comment:

And there is no discussion regarding the relationship between the substituents and the reactivities.

Response:

Thanks a lot for this comment. We explored the substrate scope bearing different substitution pattern, and found the electron-donation group and aryl group were important. The absence of one or two groups led to no reaction. We prepared two new substrates which was ineffective under standard conditions. These results were added in Fig. 2 and context.

On the other hand, there was limit on the pattern of substitution. For example, substrate **1au** and **1av** showed the electron-donating group on indene backbone and aryl ring on alkene were important as they were capable to stabilized cationic and racial intermediate.

Comment:

They proposed the generation of benzyl radical intermediate B in Figure 5, but the cyclopropyl

group used as a radical clock remained intact in product 2t. They did not explain this inconsistent result. As mentioned above, this reviewer does not recommend publication their result in Nature Communications.

Revision:

Thanks a lot for this comment. The cyclopropyl group was used as radical clock in a number of reactions to show the presence of radical. The ring opening rate is at $6.7 \times 10^7 \text{ s}^{-1}$ (*J. Org. Chem.* **1999**, *64*, 1225). We propose the rate is relatively slower than the consecutive electron transfer during the ECEC sequence. A faster clock would have more opportunity to undergo ring-opening. To verify this assumption, two new substrates **S-1a** and **S-1b** were prepared in the figure below. These substrates have only alkene as functional group, which would isolate the aziridination step from the overall formation of isoquinoline involving following steps. In substrate **S-1b**, the phenyl-substituted cyclopropyl group has a ring-opening rate at $1.5 \times 10^{11} \text{ s}^{-1}$ (*J. Am. Chem. Soc.* **1995**, *117*, 10645). The electrochemical aziridination using the standard conditions in Fig. 2 were carried out with **S-1a** and **S-1b** as starting material. We could get the aziridine **S-2a** in 54% isolated yield. **S-1b** decomposed during the reaction, giving a complex mixture. The aziridination product was not observed. After attempt, we could isolate one product. Though, the exact structure could not be identified, the cyclopropyl group disappeared as that was shown by ^1H NMR. These results suggested the radical species is highly transient and unsubstituted cyclopropyl group was kept during the transformation.

These experiment were added in supporting information:

5.2) Trapping experiment using radical clocks with different rates

Two alkenes **S-1a**² and **S-1b** were prepared in the figure below. These substrates have only alkene as functional group, which would isolate the aziridination step from the overall formation of isoquinoline involving following steps. In substrate **S-1a**, the unsubstituted cyclopropyl group has a ring-opening rate at $6.7 \times 10^7 \text{ s}^{-1}$.³ In substrate **S-1b**, the phenyl-substituted cyclopropyl group has a ring-opening rate at $1.5 \times 10^{11} \text{ s}^{-1}$.⁴ The electrochemical aziridination using the standard conditions in Fig. 2 were carried out with **S-1a** and **S-1b**. The aziridine **S-2a** was obtained in 54% isolated yield. **S-1b** decomposed during the reaction, giving a complex mixture. The aziridination product was not observed. After attempt, one product was isolate. Though, the exact structure could not be identified, the cyclopropyl group disappeared as that was shown by ^1H NMR. These results suggested the radical species is highly transient and unsubstituted cyclopropyl group was kept during the transformation.

To a flask charged with isopropyltriphenylphosphonium iodide (1.30 g, 3 mmol, 1.5 equiv) in 10 mL of anhydrous THF was added *n*-butyllithium (1.6 mL, 2.5 M in hexanes, 2 mmol, 2 equiv) at 0 °C under argon. The reaction was then stirred for 0.5 h. (4-methoxyphenyl) (2-phenylcyclopropyl) methanone (0.504 g, 2 mmol, 1 equiv) in anhydrous THF (5 mL) was added dropwise. After completion

of addition, the reaction was stirred overnight at room temperature. After being quenched with brine, the mixture was extracted with ethyl acetate for three times. The combined organic layers were washed with water, dried (MgSO₄), and concentrated. The residue was purified with flash chromatography to afford the desired product (0.334 g, 60%).

1-methoxy-4-(2-methyl-1-(2-phenylcyclopropyl)prop-1-en-1-yl)benzene (S-1a)

The **S-1a** was obtained as a white oil. ¹H NMR (400 MHz, Chloroform-*d*) δ 7.28 (t, *J* = 7.5 Hz, 2H), 7.17 (t, *J* = 7.3 Hz, 1H), 7.12 – 7.07 (m, 2H), 7.02 – 6.96 (m, 2H), 6.94 – 6.84 (m, 2H), 3.86 (s, 3H), 2.16 – 2.03 (m, 1H), 1.93 (s, 3H), 1.73 (dt, *J* = 8.9, 5.2 Hz, 1H), 1.57 (s, 3H), 1.19 – 1.04 (m, 1H), 0.84 (ddd, *J* = 8.7, 6.1, 4.8 Hz, 1H). ¹³C NMR (100 MHz, Chloroform-*d*) δ 157.8, 143.3, 135.0, 133.3, 130.8, 129.9, 128.2, 125.8, 125.3, 113.2, 55.2, 27.0, 23.6, 22.7, 20.5, 15.7.

¹H NMR (400 MHz, CDCl₃) of **S-1b**

2-cyclopropyl-2-(4-methoxyphenyl)-3,3-dimethylaziridine (**S-2a**)

Product **S-2a** was obtained as a white liquid in 54% yield (23.4 mg). ^1H NMR (400 MHz, Chloroform-*d*) δ 7.14 (d, $J = 8.7$ Hz, 2H), 6.85 (d, $J = 8.7$ Hz, 2H), 3.80 (s, 3H), 1.48 (s, 3H), 1.31 – 1.15 (m, 1H), 0.93 (s, 3H), 0.83 (s, 1H), 0.56 – 0.40 (m, 2H), 0.38 – 0.31 (m, 1H), 0.18 – 0.10 (m, 1H). ^{13}C NMR (100 MHz, Chloroform-*d*) δ 158.1, 134.4, 129.0, 113.3, 55.2, 50.3, 41.6, 24.7, 21.4, 16.0, 4.7, 3.1. HRMS m/z (ESI) calcd. for $\text{C}_{14}\text{H}_{20}\text{NO}^+$ ($\text{M} + \text{H}$) $^+$ 218.1545, found 218.1549.

¹H NMR (400 MHz, CDCl₃) of S-2a

¹³C NMR (100 MHz, CDCl₃) of S-1b

Unidentified product isolated from electrochemical reaction using S-1b

To test if the reaction proceeds via a nitrene intermediate, competition reactions employing two substrates in one cell were conducted. It was found the desired isoquinoline was obtained as normal reaction, and aziridine of other alkenes was not observed. These experiment was added to supporting information.

5.3) Competition of alkenes

To test if the reaction proceeds via a nitrene intermediate, competition reactions employing two substrates in one cell were conducted. It was found the desired isoquinoline was obtained as normal reaction, and aziridine of other alkenes was not observed.

a)

b)

Reviewer #2 (Remarks to the Author):

Liu and co-workers reported an impressive synthesis of aromatic N-heterocycles. The utilization of abundant, stable, and nitrogen-enriched ammonia directly and selectively in organic synthesis is a long-sought target, and has been shown as a highly challenging task. The activation of ammonia to nitrene or amide radical is the focused direction. Instead, this report took the other route with activation of alkene in the presence of ammonia. This transformation exploited the unique advantages of electrochemistry, such as non-acidic activation of alkene under ammonia conditions, four electron transfers in quick sequence which is so hard for other methods, and the chemoselectivity achieved by anodic oxidation converting aziridine. The protocol approaches the quantitative limit of atom economy, and meanwhile achieves a broad functional tolerance and diverse products scope. Some interesting and valuable molecules were generated in single step or with multiple electrochemical process.

Overall, this research is particularly noteworthy. So, it deserves my support to publish on Nature Communications.

There were several comments on the manuscript and supporting information:

Comment:

- 1) As the authors say, the lower yields (2ak–2ap) were partially because of the unidentified side reactions involving MeOH as nucleophile. Can sterically hindered alcohol (such as isopropyl alcohol) instead of MeOH improve the reaction?

Response:

Thanks a lot for this comment. We tried to use *i*PrOH/DCM instead of MeOH/DCM. Due to the low conductivity of *i*PrOH, the conversion of alkene was not observed. We added this result into the Table 1 and context in manuscript.

If co-solvent including *i*PrOH and DCM was applied, the conversion of alkene was not observed

due to the low conductivity (entry 9).

9ⁱ *i*PrOH/DCM GF+ / Ag- Mg(ClO₄)₂ N. R.

Comment:

2) NH₃ gas was directly introduced into the reaction. Does the concentration of NH₃ affect the reaction efficiency?

Response:

Thanks a lot for this comment. We carried out experiment using different concentration of ammonia. The results were listed below, and it was found a 0.28 M ammonia solution in MeOH/DCM was adequate to give comparable yields.

	NH ₃ (7 mol/L in MeOH)	Solvent	Yield (%) ^c
1 ^a	0.7 mmol (0.1 mL)	CH ₃ OH (3.9 mL)+DCM (1.0 mL)	54
2 ^a	1.4 mmol (1 mL)	CH ₃ OH (3.0 mL)+DCM (1.0 mL)	64
3 ^a	28.0 mmol (4 mL)	DCM (1.0 mL)	67

One entry was added to manuscript in Table 1 and context.

Instead of ammonia atmosphere, a solution of ammonia of 0.28 mol/L was adequate to give comparable yield (entry 10).

10^j CH₃OH/DCM GF+/Ag- Mg(ClO₄)₂ 64

Comment:

3) Aqueous ammonia would be readily to handle and store. How about the result of NH₄OH in this reaction?

Response:

Thanks a lot for this comment about one readily alternative for gaseous ammonia. We carried out the reaction using NH₄OH, the desired product was obtained in 10% yield with other side products.

Along with data of the above table, these results of ammonia source were added to Supporting Information section 2:

Optimization of ammonia source

Entry	NH ₃ source	Solvent	Yield (%) ^b
1 ^a	NH ₃ (7 mol/L in MeOH) 0.7 mmol (0.1 mL)	CH ₃ OH (3.9 mL)+DCM (1.0 mL)	54
2 ^a	NH ₃ (7 mol/L in MeOH) 1.4 mmol (1 mL)	CH ₃ OH (3.0 mL)+DCM (1.0 mL)	64
3 ^a	NH ₃ (7 mol/L in MeOH) 28.0 mmol (4 mL)	DCM (1.0 mL)	67
4	aq. NH ₄ OH (4 mL)	DCM	10

^a Reaction conditions: **1a** (0.1 mmol), Mg(ClO₄)₂ (0.1 mmol), solvent (5.0 mL), 4.5 V cell potential, rt, Ar, 3 h. ^b ¹H NMR yields.

Comment:

4) The discussion on the synthesis of **2ar** need additional information.

Response:

Thanks a lot for this comment. We added the discussion of the reaction in Fig.3 and context in manuscript:

By employing Hantzsch ester (**3**) as starting material, a tandem electrochemical dehydrogenative oxidation/reductive ring contraction/NH₃ insertion, gave the 1,3-diazine (**2ar**) without isolation of intermediates.

a

An intermediate was used as substrate to conduct the NH₃ insertion, corresponding product was obtained in 38% yield. This information was added in supporting information in section 10.

10. The electrochemical synthesis of diazine from pyrrole **5**

5 was prepared according to the known procedure¹³. Following the general procedure A, a 10 mL three-necked heart-shaped flask was charged with **5** (0.1 mmol), Mg(ClO₄)₂ (0.1 mmol) and a magnetic stir bar. The flask was equipped with a rubber stopper, graphite felt (2 cm x 1 cm x 0.5 cm) as anode and Ag plate (2 cm x 1 cm) as cathode. The flask was evacuated and backfilled with ammonia gas for three times, then an ammonia gas balloon was connected to this flask via a needle. Next, 3 mL of anhydrous MeOH and 2 mL of anhydrous DCM was added via syringe. The electrolysis with 4.5 V cell potential was carried out at room temperature. After 4 hours, the mixture was concentrated under reduced pressure. The residue was purified by chromatography on silica gel to afford the desired product **2ar** (9 mg, 38%).

Reviewer #3 (Remarks to the Author):

In this communication, Chen and their co-worker reported a rare example to construct aromatic N-heterocycles via insertion of ammonia. This process was enabled via electrochemical aziridination and ring expansion. This operationally straightforward proved useful for the preparation of versatile heterocycles and pharmaceutical compounds in good yields. The results are promising, and the authors showcase the potential for further applications. This manuscript has its own merit in the employment of electrosynthesis, demonstrating various aromatic N-heterocycles with good yields in a concise manner, and this work will catch the attention of a broad readership. To sum up, I think that this is a nice piece of work well suited to be published in Nature Communication after revising the items noted below:

Comment:

1) The ammonia gas pressure used in the reaction should be added to the table footnote.

Response:

Thanks a lot for this comment. This parameter is important involving a gaseous phase reactant. We added the pressure of ammonia in Table 1 and Fig.2 as:

NH₃ (balloon, ca. 1 atm)

Comment:

2) Given the full name of Fg and GF.

Response:

Thanks a lot for this comment. The full name and abbreviation was correlated at the first place when appeared in the manuscript.

The first observed insertion product **2a** was observed with a 16% ¹H NMR yield when graphite felt (GF) was used as electrodes in methanol (entry 1).
oxidation-labile functional group (Fg) tolerance, and unexplored selectivity (Fig. 1b).

Comment:

3) ¹H NMR of **1m** was not clean and should be improved.

Response:

Thanks a lot for this comment. The compound **1m** was purified to improved ¹H NMR spectrum. The new spectrum was added to the Supplementary file.

¹H NMR (400 MHz, CDCl₃) of **1m**

Comment:

4) Kinetic studies of intermediate aziridine **9** and the product were suggested to be investigated.

Response:

Thanks a lot for this important comment. We carried out kinetic study on the intermediate **9** and product **2a**. As the intermediate **9** was too unstable to get its' Jobs plot, we used the ratio of peak area of **9** to internal standard (*n*C₁₂H₂₆) on GCMS instead of exact concentration of **9**. It was found the concentration of **9** reaches the maximum at 30 minutes, and start to decrease afterwards. The concentration of product reaches the maximum at about 120 minute and dropped a little bit at 180 minutes. These results suggest careful monitor of reaction progress is important to achieve the optimum yield. The kinetic profiles of **9** and **2a** were added in Fig. 4a and context in manuscript.

In the kinetic study, the concentration of intermediate **9** reached the maximum at 30 minutes, and then decreased to full consumption until 180 minutes. In comparison, product **2a** generated continuously until 120 minutes, and the yield dropped a little bit afterwards (Fig. 4a).

The details of kinetic study were added in Supplementary Information, section 5.

Comment:

- 5) In Fig. 3a, a nice electrochemical sequence for the synthesis of 1,3-diazine 2a was reported. The intermediates should also be added to the manuscript.

Response:

Thanks a lot for this comment. The intermediates were added in the Fig. 3a and context.

By employing Hantzsch ester (**3**) as starting material, a tandem electrochemical oxidation/ring contraction/NH₃ insertion, gave the 1,3-diazine (**2a**) without isolation of intermediates.

An intermediate was used as substrate to conduct the NH_3 insertion, corresponding product was obtained in 38% yield. This information was added in supporting information in section 10.

10. The electrochemical synthesis of diazine from pyrrole S-3

S-3 was prepared according to the known procedure¹³. Following the general procedure A, a 10 mL three-necked heart-shaped flask was charged with **S-3** (0.1 mmol), $\text{Mg}(\text{ClO}_4)_2$ (0.1 mmol) and a magnetic stir bar. The flask was equipped with a rubber stopper, graphite felt (2 cm x 1 cm x 0.5 cm) as anode and Ag plate (2 cm x 1 cm) as cathode. The flask was evacuated and backfilled with ammonia gas

for three times, then an ammonia gas balloon was connected to this flask via a needle. Next, 3 mL of anhydrous MeOH and 2 mL of anhydrous DCM was added via syringe. The electrolysis with 4.5 V cell potential was carried out at room temperature. After 4 hours, the mixture was concentrated under reduced pressure. The residue was purified by chromatography on silica gel to afford the desired product **2ar** (9 mg, 38%).

Comment:

Furthermore, could substitute pyrrole be the suitable substrate in the transformation?

Response:

We tried several substrate including pyrrole moiety, but failed to get the corresponding products. The reaction gave complex mixtures. We added these examples in Supplementary Information, section 14.

14. Substrates giving complex mixtures

** See Nature Research's author and referees' website at www.nature.com/authors for information about policies, services and author benefits.

Our flexible approach during the COVID-19 pandemic

If you need more time at any stage of the peer-review process, please do let us know. While our systems will continue to remind you of the original timelines, we aim to be as flexible as possible during the current pandemic.

This email has been sent through the Springer Nature Tracking System NY-610A-NPG&MTS

Confidentiality Statement:

This e-mail is confidential and subject to copyright. Any unauthorised use or disclosure of its contents is prohibited. If you have received this email in error please notify our Manuscript Tracking System Helpdesk team at <http://platformsupport.nature.com>.

Details of the confidentiality and pre-publicity policy may be found here <http://www.nature.com/authors/policies/confidentiality.html>

Privacy Policy | Update Profile

REVIEWERS' COMMENTS

Reviewer #1 (Remarks to the Author):

These authors have revised their manuscript according to the comments. This reviewer recommends publication of their revised manuscript as is.

Reviewer #2 (Remarks to the Author):

In this revised manuscript authors have addressed the referee comments and the manuscript has been improved. Now this revised version of the manuscript may be accepted in its current form.

Reviewer #3 (Remarks to the Author):

Based on the comments pointed out by this reviewer, the authors of this paper have substantially revised the original manuscript. Thus, this reviewer recommends its publication as it stands.